# SARS-CoV-2 Seroprevalence in Unvaccinated Adults in Thailand in November 2021

**DOI:** 10.3390/vaccines10122169

**Published:** 2022-12-16

**Authors:** Surakameth Mahasirimongkol, Ballang Uppapong, Wiroj Puangtubtim, Panadda Dhepakson, Parnuphan Panyajai, Naphatcha Thawong, Nadthanan Pinyosukhee, Archawin Rojanawiwat, Nuanjun Wichukchinda, Sakulrat Soonthorncharttrawat, Kanisorn Larpardisorn, Sumet Amonyingcharoen, Kritchai Juntaped, Tassanee Chaiyakum, Chayada Tongkamsen, Jeerapa Srilaket, Jiratikamon Chipatoom, Rattanawadee Wichajarn, Nutchanat Chatchawankanpanich, Lapasrada Pattarapreeyakul, Porntip Chaiya, Kaveewan Mongkolsiri, Suthida Tuntigumthon, Kritsamon Sophondilok, Nalinee Saengtong, Kodcharad Jongpitisub, Supakit Sirilak

**Affiliations:** 1Medical Life Sciences Institute, Department of Medical Sciences, Ministry of Public Health, Nonthaburi 11000, Thailand; 2Administrative Offices, Department of Medical Sciences, Ministry of Public Health, Nonthaburi 11000, Thailand; 3National Institute of Health, Department of Medical Sciences, Ministry of Public Health, Nonthaburi 11000, Thailand; 4Medical Sciences Technical Office, Department of Medical Sciences, Ministry of Public Health, Nonthaburi 11000, Thailand

**Keywords:** seroprevalence, SARS-CoV-2, COVID-19, antibody, anti-S, unvaccinated, Thailand, 2021

## Abstract

Between the first case of COVID-19 in January 2020 and the end of 2021, Thailand experienced four waves of the epidemic. The third and fourth waves were caused by the alpha and delta strains from April 2021 to November 2021. Serosurveillance studies provide snapshots of the true scale of the outbreak, including the asymptomatic infections that could not be fully captured by a hospital-based case detection system. We aimed to investigate the distribution of SARs-CoV-2 seroprevalence in unvaccinated adults after the delta wave outbreak. From November to December 2021, we conducted a cross-sectional survey study in 12 public health areas (PHAs) across Thailand. A total of 26,717 blood samples were collected and tested for SARs-CoV-2 antibodies (anti-S IgG) using a qualitative immunoassay. The results showed that seropositive prevalence in this cohort was 1.4% (95% CI: 1.24 to 1.52). The lowest prevalence was in the northern region (PHA 1) and in central Thailand (PHA 3) at 0.4% (95% CI: 0.15 to 0.95), while the highest was in the southern region of Thailand (PHA 12) at 5.8% (95% CI: 4.48 to 7.29). This seropositive prevalence was strikingly lower than the reports from other countries. Our serosurveillance results suggest that the vaccination of unvaccinated groups should be accelerated, especially in the public health areas with the lowest seroprevalence.

## 1. Introduction

The coronavirus disease 2019 (COVID-19) is an infectious disease caused by severe acute respiratory syndrome coronavirus 2 (SARS-CoV-2) [1,2]. This illness led to globally catastrophic outcomes, resulting in more than four million deaths as of September 2021 [3]. In Thailand, the first wave of COVID-19 hit in late March 2020, and the fourth wave began in the form of the delta variant during mid-2021 [4]. As a result, public health measures and movement restrictions, such as lockdowns, school closures, working from home, and social distancing, were implemented to control the transmission of the disease.

Several techniques are employed for COVID-19 diagnosis. Considering the breakthroughs in medical diagnosis, nucleic-acid-based approaches are destined to become a rapid and reliable method. Among these, real-time reverse-transcriptase PCR (real-time RT-PCR) is preferred as a gold standard due to its advantages as a specific and simple qualitative assay [5]. Samples from the upper respiratory tract, including nasopharyngeal swabs, can be tested using real-time RT-PCR.

The World Health Organization (WHO) estimated that around 80% of COVID-19 cases are asymptomatic or mild, 15% are severe, and 5% are critical, requiring a ventilator [6]. Pre-symptomatic or asymptomatic cases have a prolonged period of viral shedding and still transmit infections [7,8]; unfortunately, these cases do not undergo real-time RT-PCR testing. Therefore, the infection rate reported based on real-time RT-PCR testing of the population cannot fully reflect the COVID-19 outbreak situation. To investigate the true nature of an outbreak, serological surveys should be implemented to assess the population infected with SARS-CoV-2, including asymptomatic and symptomatic cases [9].

In Thailand, serological studies were conducted before the delta wave. A study among non-healthcare workers with a high potential of exposure to SARS-CoV-2 in the northern regions (the Chiang Mai and Lamphun provinces) during the second wave of the outbreak (November 2020–January 2021) reported 0.9% (*n* = 1651) positive results for antibody tests [10]. Another study in healthcare workers in the Bangkok metropolitan area, and the western and eastern regions of Thailand from January to March 2021 reported that the positive rate for SARS-Cov-2 IgG-spike antibodies was 0.2% (*n* = 600) [11]. However, no data from serological survey studies conducted after the outbreak of the fourth wave in Thailand are available, while several countries reported various prevalence rates of COVID-19 seropositivity after the delta outbreak. In the United States, the seroprevalence based on blood donation testing reached 94.7% (*n* = 2,408,093) by December 2021 [12]. In Canada, the antibody-positive rate among blood donors between November 13 and 24, 2021 was 100% (*n* = 9018) [13]. In Scotland, the seroprevalence in people attending community healthcare settings from 15 November to 19 December 2021 was estimated to be 87.4% (*n* = 2816) [14]. In South Africa, a study on blood donors in eight provinces from 8 to 12 November 2021 found that the antibody-positive rate was about 71.1% (*n* = 3395) [15].

To investigate the true number of COVID-19 infection cases after the delta outbreak in Thailand, a study was performed among unvaccinated Thais to determine SARS-CoV-2 seroprevalence between November and December 2021. This survey, by determining the prevalence of natural infections, could aid the government in preparing effective strategies for COVID-19 prevention in the next waves of the pandemic.

## 2. Materials and Methods

### 2.1. Study Design and Setting

From November to December 2021, we conducted a cross-sectional survey study in 12 public health areas across Thailand. We approached subjects who visited COVID-19 vaccination centers regarding participation in this project. Enrolled subjects had to be aged between 18 and 60 years old, with no history of COVID-19 illness or vaccination. After providing written informed consent, the participants were bled and interviewed to obtain information on gender, occupation, place of residence, and number of household members. SARs-CoV-2 antibody status tests were administered to all the subjects individually.

### 2.2. Sampling Plan Strategies

People were sampled after the 4th pandemic wave in Thailand. Apparently, this was performed rather proportionally across the 13 health–administrative Thai regions. Subjects were enrolled based on voluntary participation at vaccination points: 40,694 people coming in were informed to participate in the project; in addition, requests for personnel data and blood donations were put forward if they met the basic criterion of not being vaccinated up to the point of participation, and not being clinically ill to the best of their knowledge with COVID-19. The summary for the sampling plane is show in Figure 1.

### 2.3. Blood Collection and Processing

Blood samples (3–5 mL) were separated into serum samples with cold chains, and aliquots were placed into two microtubes: one for SARs-CoV-2 antibody testing and another for back up. All the specimens were kept at −20 °C until laboratory testing was performed.

### 2.4. Laboratory Testing

#### 2.4.1. SARS-CoV-2 Antibody Testing by In-House Immunoassay (ELISA)

In-house immunoassay testing was conducted at the laboratory of Medical Life Sciences Institute. The ELISA microplate (Thermo Scientific, Roskilde, Denmark) was coated over-night at 4 °C with 0.02 ug of SARS-CoV-2 spike RBD proteins per well and further washed with 400 µL of wash buffer (PBST; 0.02% in phosphate-buffered saline) 3 times. Then, the ELISA microplate was blocked with 300 µL/well of blocking buffer (5% skim milk in PBST) at ambient temperature for 1 h; then, 50 µL/well of the test and control serum (1:200 dilution) was added into the microplate, and incubated at ambient temperatures for 1 h. After 3-time washing, 50 µL of 1:10,000 diluted polyclonal rabbit anti-human IgG conjugated HRP (Dako, cat no. P0214) was added. The microplate was incubated for 1 h at ambient temperatures and washed 5 times. After that, 50 µL of TMB peroxidase substrate (SeraCare, Milford, MA, USA) was added and further incubated in the dark at ambient temperatures for 30 min. The reaction was stopped by adding 50 µL of 1 M H_2_SO_4_, and the optical density (OD) was read at 450/570 nm with an ELISA microplate reader (TECAN, Männedorf, Switzerland). Finally, the OD ratio was calculated by dividing the OD of the sample by the OD of the negative control.

The performance of the in-house ELISA was evaluated by testing with 173 confirmed positive samples and 228 confirmed negative samples, and compared to the commercial assay (Quant IgG II, Abbott Ireland, Sligo, Ireland). Using a cut-off value of 1.5, which is determined as the mean values of OD ratio derived from confirmed negative samples plus two standard deviations, demonstrated a sensitivity and specificity of 100% and 95.61%, respectively. The positive samples were further confirmed by the commercial quantitative test.

#### 2.4.2. Quantitative for SARS-CoV-2

Individual serum was quantified for SARS-CoV-2 antibodies relative to S1 SARS-CoV-2 subunit spike proteins by a commercial assay (Quant IgGII, Abbott Ireland, Sligo, Ireland), which is a chemiluminescent microparticle immunoassay (CMIA), run on automatic analyzer with the ARCHITECT I System (Abbott, Abbott Park, IL, USA). The reportable range of this kit was 6.8–80,000 Abbott Arbitrary Unit (AU/mL). The correlation level of the antibody when compared with WHO’s International Standard (NIBSC code 20-136) can be converted into a binding antibody unit (BAU/mL) by multiplying 0.142 at the 0.999 correlation level. The sample, which has over 50 AU/mL, is considered positive for SARS-CoV-2 antibodies.

### 2.5. Data Collection and Data Analysis

Individual data were collected in case record form (CRF); we collected the participants’ characteristics data, such as gender, age, occupation, subject’s residence area (public health area (PH) 1 to 12), body mass index, underlying disease, the number of people the subject lived with, and close exposure to COVID-19 cases.

## 3. Results

### 3.1. Participant Data

From November to December 2021, 40,694 people were asked to participate in the project based on voluntary participation, and they met the basic criterion of not being vaccinated up until that point and not being clinically ill. Participants numbering 26,783 were enrolled; however, those providing consent and who were willing to donate blood samples resulted in a final number of 26,717 participants. The median age of participants was 31 years (IQR: 25–50); 51.9% were female; 53.2% were within the normal body mass index; most (79.4%) did not have underlying diseases; 64.7% had a non-salary base occupation; 55.6% stayed with 3–5 persons in the same house; and only 1.7% came in close contact with COVID-19 cases (the data on the participant’s characteristics are shown in Table 1). The number of participants in each public health area ranged from 1084 to 4084, as shown in Table 2.

### 3.2. Serological Test Result

A total of 26,717 blood samples were interpreted as serological test results, as shown in the flowchart in Figure 2; 367 clinical specimens tested positive for anti-SARS-CoV-2 antibodies, which indicates a 1.4% seroprevalence in this investigation (Table 2). The distribution of seropositive results and the data on participant characteristics are shown in Table 2 and Table 3, respectively. Public health area 1 in the northern region and public health area 3 in the central region indicate the lowest seropositive cases at 0.4% (95% CI: 0.15 to 0.95), while public health area 12 in the southern region hits the highest seropositive peak at 5.8% (95% CI: 4.48 to 7.29).

### 3.3. Seroprevalence by Geographic Data

This survey study employed samples from Thai people who visited vaccination points and those that voluntarily agreed to participate in the cross-sectional study of epidemiological analyses, as shown in Figure 3. The 12 public health areas are categorized by geographic region: northern, northeastern, central, and southern regions. To begin with, in the northern region, clinical specimens were collected from four provinces from a total of eight provinces in public health area 1, including Chaing Rai, Lampang, Phayao, and Lamphun; in addition, the results show a 0.4% seropositive result (6/1416), whereas Phetchabun and Pitsanulok from public health area 2 have four-times-higher seropositive results at 1.6% (24/1431). The central region consists of public health area 3, 4, 5, and 6. The lowest seropositive results belong to public health area 3 at 0.4% (6/1366), with collected specimens from Nakhon sawan and Uthai Thani. In contrast, public health areas 4, 5, and 6 demonstrated results of 2.5% (34/1382), 2.7% (29/1084), and 2.8% (70/2517), respectively. Public health areas 7, 8, 9, and 10 are located in the northeastern region; and their seropositive rates are 1.1% (40/3726), 0.9% (27/3137), 0.9% (35/4084), and 0.5% (17/3702), respectively. Furthermore, the southern zone comprises public health areas 11 and 12. Public health area 11 shows 0.8% in terms of seropositivity results. Distinctly, public health area 12 displays the greatest seroprevalence rate at 5.8%.

## 4. Discussion

This cross-sectional survey aimed to study the seroprevalence of IgG SARS-CoV-2 antibodies in order to estimate the burden of infections during the late-fourth-waves of the pandemic in Thailand (November–December 2021). This period of time shows the decreasing number of confirmed COVID-19 cases (approximately 5000 cases/day) compared to the highest infection of the fourth wave in August 2021 (approximately 20,000 cases/day) [3]. In this study, serum samples were collected from unvaccinated donors who did not have historical reports of SARS-CoV-2 infections: 26,717 persons. These participants should be illustrative of the naïve immunity that Thai subjects have; however, seropositive results in these groups are characterized as asymptomatic COVID-19 patients. We found a seroprevalence of 1.4% in the studies. This prevalence increases from the beginning of the fourth wave (April 2021) with 0.2% seropositive results [11,16].

There were only minor differences in seropositivity between sexes, age groups, and BMI categories. However, seroprevalence in women was slightly higher than men. The seroprevalence in age groups ranged from 1.2 to 1.6%. The lowest rate of seropositivity was observed in the 18–27 age group and the highest rate was in the 28–37 age group. These findings can be partially explained by sociological data from a national survey and are related to the highest number of COVID-19 patients in the same age group during the fourth wave (April–May 2021) [17]. The highest seropositivity prevalence was observed in persons with BMI > 30 kg/m^2^. This result suggested that BMI may be related to SARS-CoV-2 infections; moreover, there are supporting data from UK that BMI is related to SAR-CoV-2 seropositive results and COVID-19-related death [18].

In Thailand, the ministry of public health established 13 health areas for public health management. This governing structure is becoming more prominent during responses to the pandemic. The health regions with a high number of tourism activity or immigrant workers are susceptible to COVID-19 outbreaks, and reflected in the better responses observed in disease control. Health regions with exposure to border areas include health regions 1, 2, 4, and 5 in the north of Thailand, which are located next to Myanmar; health region 6, located next to Cambodia; and health region 12, located next to Malaysia. These health regions are the primary areas responding to the initial clusters of COVID-19. At the end of 2020, immigrant worker outbreaks were detected in health region 5, where immigrant workers from Myanmar were displaced from their countries. As a result of immigrant workers coming from Cambodia in mid-2021, the SARS-CoV-2 delta variant was introduced in Thailand. Afterward, the delta variants spread from Bangkok across the country during the latter half of 2021. On the other hand, the SARS-CoV-2 beta variant circulated concomitantly in health region 12, which is extended from Malaysia. With mixed SARS-CoV-2 variants, the seropositivity rate in various health regions supported the epidemiology, while the immunoassay used in this study could not discriminate the effects of various strains due to the non-specificity of the anti-S against various SARS-CoV-2 variants of concern (VOCs). The detailed neutralization-based assay may be able to describe more details with respect to the increasing viral infections and immune escapes [19,20].

However, Bangkok, which inhibits public health area 13, was not included in this serosurveillance study because a number of unvaccinated people were in the minor age group. In addition, Bangkok had a 98.5% vaccination coverage and a significant number of SARS-CoV-2-infected cases. As a result, estimating seroprevalences in the Bangkok region was insufficient, and vaccination may have resulted in a larger true prevalence of seropositivity.

## 5. Conclusions

After the fourth wave of the COVID-19 pandemic in Thailand (November–December 2021), our study estimated that seroprevalence is around 58.7% for unvaccinated Thai subjects (illustrated for asymptomatic cases), confirmatory cases by RT-PCR (illustrated for infected cases), and vaccination (illustrated for immunized cases), which shows that Thai subjects had seroprevalence rates that are lower than those of some developed countries; for example, 83% seroprevalence is observed in the United States of America and 89% is observed in Germany [20,21]. The advantage of our study is that it is the largest study of the seroprevalence of anti-SARS-CoV-2 IgG antibodies in Thailand. According to the WHO population-based age-stratified seroepidemiological investigation protocol, the major strengths of our study include size and coverage. Confirmed COVID-19 patients comprise 2.6% of the Thai population; meanwhile, asymptomatic patients that are not included in the detection system comprise around 1.4%.

## Figures and Tables

**Figure 1 vaccines-10-02169-f001:**
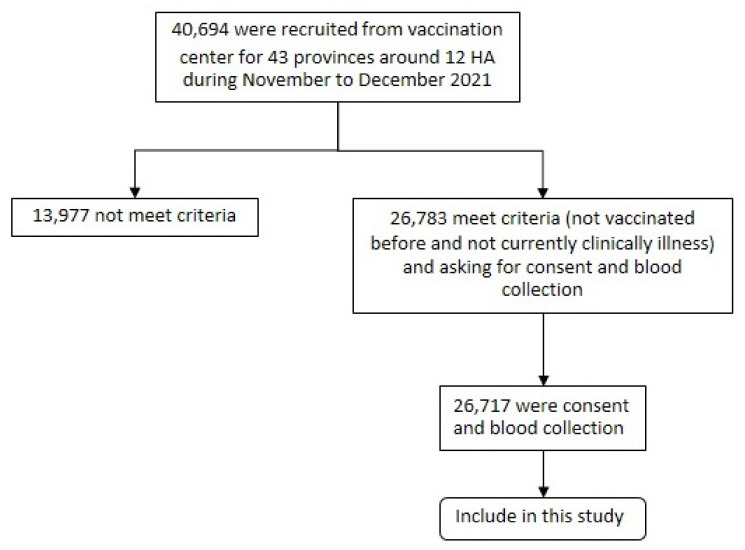
The sampling strategy: in total, 40,694 were enrolled based on voluntary participation at vaccination center. Only 26,783 met the criteria and were asked to donate a blood sample.

**Figure 2 vaccines-10-02169-f002:**
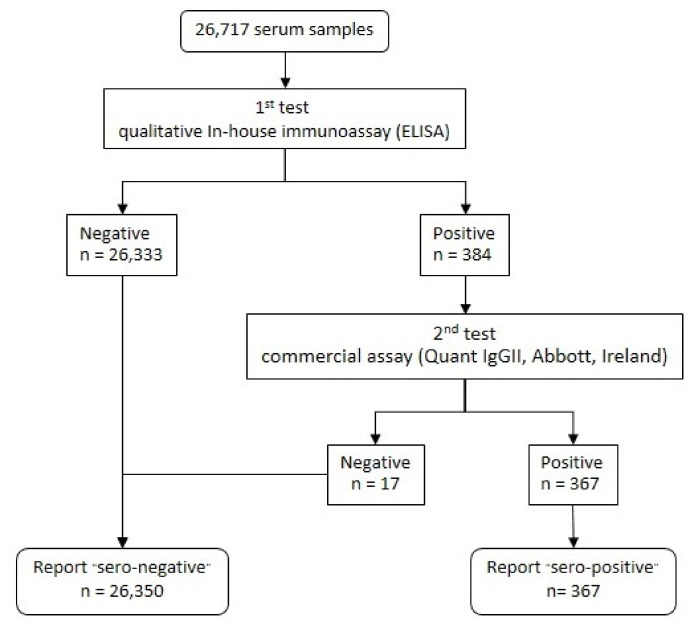
A flowchart of the interpreted sample results of the 1st and 2nd SARS-CoV-2 antibody tests.

**Figure 3 vaccines-10-02169-f003:**
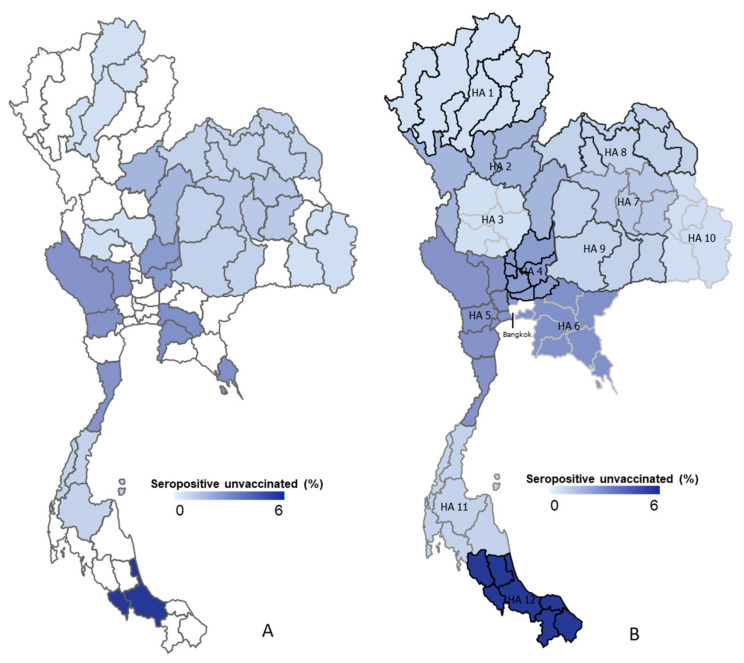
Mapping 13 public health areas and seropositive distributions in Thailand. (**A**) The random province sites around Thailand that show each HA (see Appendix A Table A1). (**B**) The density of seropositivity in each area of public health, which ranges from 0% to 6% seropositivity, is demonstrated by a choropleth map. HA, health area.

**Table 1 vaccines-10-02169-t001:** Participant characteristics.

Characteristics	Count	Percentage
Gender		
Female	13,873	51.9
Male	12,844	48.1
Age groups (years)		
18–27	8158	30.5
28–37	5186	19.4
38–47	5146	19.3
48–57	6127	22.9
≥58	2100	7.9
Body mass index (BMI) groups (kg/m^2^)		
<18.5 (underweight)	2564	9.6
18.5–24.9	14,221	53.2
25–30 (overweight)	6240	23.4
>30 (obese)	2593	9.7
Not defined	1099	11.1
Blood pressure (SBP/DBP)		
Normal (91–120/61–80)	7144	26.8
Pre high (121–140/81–90)	5873	21.9
High (141–190/91–100)	2674	10.0
Not defined	11,026	41.3
Occupation groups		
Government employed (e.g., military, police)	1282	4.8
Non-government employed (e.g., clerk, accountant)	846	3.2
Freelance (non-salaried worker, owner, farmer, trader, rider)	17,309	64.8
Other (e.g., student, monk, retired, unemployed)	2898	10.8
Not defined	4382	16.4
Number of household member		
Alone	3644	13.6
1–2	4089	15.3
3–5	14,868	55.6
>5	4116	15.4
Underlying diseases		
No	21,210	79.4
Yes	5507	20.6
Close contact with COVID-19 cases		
No	26,275	98.3
Yes	442	1.7

Demographic characteristics data of participants in the final sample of 26,717 participants.

**Table 2 vaccines-10-02169-t002:** Percentage of seropositive participants by the geography of Thailand.

Public Health Areas	*n*	SARS-CoV-2 IgG Seropositive (Cases)	% Seropositive (95% CI)
North			
Public health area 1	1416	6	0.4 (0.16–0.92)
Public health area 2	1431	24	1.6 (1.08–2.49)
Central			
Public health area 3	1366	6	0.4 (0.16–0.95)
Public health area 4	1382	34	2.5 (1.71–3.42)
Public health area 5	1084	29	2.7 (1.80–3.82)
Public health area 6	2517	70	2.8 (2.17–3.50)
North-East			
Public health area 7	3726	40	1.1 (0.77–1.46)
Public health area 8	3137	27	0.9 (0.57–1.25)
Public health area 9	4084	35	0.9 (0.60–1.19)
Public health area 10	3702	17	0.5 (0.27–0.73)
South			
Public health area 11	1744	14	0.8 (0.44–1.34)
Public health area 12	1128	65	5.8 (4.48–7.29)
Bangkok (Public health area 13)	-	-	-
**Overall**	**26,717**	**367**	**1.4 (1.24–1.52)**

**Table 3 vaccines-10-02169-t003:** SARS-CoV-2 IgG serological test result.

Characteristics	Count	SARS-CoV-2 IgG Seropositive (Cases)	% Seropositive (95% CI)
Gender			
Female	13,873	219	1.6 (1.38–1.80)
Male	12,844	148	1.2 (0.98–1.35)
Age groups (years)			
18–27	8150	101	1.2 (0.10–1.50)
28–37	5186	82	1.6 (1.26–1.96)
38–47	5146	79	1.5 (1.22–1.91)
48–57	6127	71	1.2 (0.91–1.46)
≥58	2100	33	1.6 (1.08–2.19)
Body Mass Index (BMI) groups (kg/m^2^)			
<18.5 (underweight)	2564	32	1.2 (0.86–1.76)
18.5–24.9	14,221	172	1.2 (1.04–1.40)
25–30 (overweight)	6240	90	1.4 (1.16–1.77)
>30 (obese)	2593	52	2.0 (1.50–2.62)
Not defined	1099	21	1.9 (1.19–2.9.1)
Blood Pressure (SBP/DBP)			
Normal (70–120/40–80)	7144	77	1.1 (0.85–1.34)
Pre high (121–140/81–90)	5873	75	1.3 (1.01–1.60)
High (141–190/91–100)	2674	22	0.8 (0.51–1.24)
Not defined	11,026	193	1.8 (1.51–2.01)
Occupations groups			
Government employed (e.g., military, police)	1282	20	1.6 (0.96–2.40)
Non-government employed (e.g., clerk, accountant)	846	12	1.4 (0.74–2.46)
Freelance (non-salaried worker, owner, farmer, trader, rider)	17,309	235	1.4 (1.19–1.54)
Other (e.g., student, monk, retired, unemployed)	2898	20	0.7 (0.42–1.06)
Not defined	4382	80	1.8 (1.45–2.27)
Number of household member			
Alone	3644	72	1.9 (1.55–2.48)
1–2	4089	51	1.2 (0.93–1.64)
3–5	14,868	178	1.2 (1.03–1.39)
>5	4116	66	1.6 (1.24–2.04)
Underlying Diseases			
No	21,210	291	1.4 (1.22–1.54)
Yes	5507	76	1.4 (1.09–1.72)
Close Contact with COVID-19 cases			
No	26,275	330	1.3 (1.12–1.40)
Yes	442	37	8.3 (5.96–11.35)

## Data Availability

Not applicable.

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
