# Peer review of "SARS-CoV-2 Seroprevalence in Unvaccinated Adults in Thailand in November 2021"

_vaccines, 2022, doi:10.3390/vaccines10122169_

Round 1

Reviewer 1 Report

Manuscript reports on prevalence of Sars-CoV2 anti-S IgG positive people in a sample Thai subjects who (based on their own report) before the testing had not suffered clnically manifest COVID-19 disease and were not vaccinated. People were sampled after the 4th pandemic wave in Thailand. Apparently, this was done rather proportionally across the 13 health-administrative Thai regions. Subjects were enrolled based on voluntary participation - at vaccination points: people comming in were asked to donate a blood sample (if they met the basic criterion of not beeing vaccinated up to that point and not beeing clinically ill - to the best of their knowledge - with COVID-19). Reports on COVID-19 seroprevalence from different countries/parts of the World are informative. However, there are several limitations to the present manuscript.

Comments

1. English is not my native language - but it is rather obvious that this manuscript is heavily burdened with grammatical and spelling errors and typing errors. Extensive English editing is needed.

2. At one point, authors refer to the process of subject inclusion as "simple random sampling". But this is not true - the included people are NOT a "(simple) random sample", nor was this technique employed. These are people who visited vaccination points and voluntarily agreed to participate - a  convenient sample.

3. Seeminlgy, the tendecy was to include people from all administrative regions - which apparently was successfully done. But authors should in more detail describe the sampling strategy. E.g., how did they define the total number to include and how did they (if they did!) conclude on the number of subjects to be enrolled by an administrative region.

4. Hence, while the people included in this survey indeed were most likely (e.g., accounting on a minimal recall bias) "naive" in terms of no clinically manifest disease/not vaccianted - the sampe is NOT representative. This does not reduce the value of the work - but the fact is, that participants may be defined as "illustrative of" ..but not representative.

5. It should also be clearly depicted why was there an upper limit of age. Adults (18+ subjects) were included, but justification for the upper age limit is needed.

6. It is impossible to conclude on "risk factors" in a cross-sectional design. At best, one might speak about "associations" (between certain subject characteristics and the fact of being seropositive) - but, when you only know 3-4 variables about the subjects, this makes no sense - if you only know age, gender, number of household members and broadly classified occupation - you cannot detect "independent association". And this analysis really does not add anything to this work . So, besides being conceptually wrong, it is non-informative - and should be removed.

7. Obviously, it would have been too expesive to conduct all 25000+ tests using a commercial test. Therefore, authors developed an in-house immunoassay. This might be the "critical point" of the entire work. Method describes the technical part - but many other things have to be declared about this assay: (i) all analyses done at one place/lab? (ii) validation data for the assay ? (eg., "agreement" vs. a commercial one, intra-day and inter-day variability etc.). This is the critical bioanalytical procedure in this work - everything depends on it. Hence, it needs to be fully validated and this should be presented in the manuscript (or a Supplement). 

8. A flow-chart should show how many tested positive in the "first-step" testing (in-hous assay) - i.e., how many were conveyed to further testing using a commercial assay (to yield the final "positives").

9. Discussion should avoid wording like "representative sample"..."random sample.." etc. It is true that subjects were enrolled without a "general prejudice" - they just needed to meet the key inclusion criterion and consent to provide a blood sample, but this does not mean that they are "random sample from the population".

10. Proportions of seroprevalent should not be interpreted in terms of "differences between regions" - but only in terms of "differences between samples from different regions". The sampling process is not appropriate for generalizations - hence, authors should be cautions in this respect.

11. Overall, there are several key points that need to be clearly presented: (i) details of the sampling plan; (ii) validation data for the in-house assay; (iii) interpretation in terms of the "sample" -with very cautions reference to the "population of Thai adults who had not been vaccinated and/or had not experienced a clinically obvious COVID-19 episode".

12. Authors should reconsider what is a "broader" interpretation of the findings. Do they really indicate what the last sentence of the Conclusions states? 

Author Response

Response to Reviewer 1 Comments

Point 1: English is not my native language - but it is rather obvious that this manuscript is heavily burdened with grammatical and spelling errors and typing errors. Extensive English editing is needed.

Response 1: We will submit this journal's English editing after peer review.

Point 2: At one point, authors refer to the process of subject inclusion as "simple random sampling". But this is not true - the included people are NOT a "(simple) random sample", nor was this technique employed. These are people who visited vaccination points and voluntarily agreed to participate – a convenient sample.

Response 2: we changed the word from "random sample" to "convenient sample"

Point 3: Seeminlgy, the tendecy was to include people from all administrative regions - which apparently was successfully done. But authors should in more detail describe the sampling strategy. E.g., how did they define the total number to include and how did they (if they did!) conclude on the number of subjects to be enrolled by an administrative region.

Response 3: we added 2.2 sampling plane strategies.

Point 4: Hence, while the people included in this survey indeed were most likely (e.g., accounting on a minimal recall bias) "naive" in terms of no clinically manifest disease/not vaccinated - the sample is NOT representative. This does not reduce the value of the work - but the fact is, that participants may be defined as "illustrative of" but not representative.

Response 4: we changed the word from "representative" to "illustrative".

Point 5: It should also be clearly depicted why was there an upper limit of age. Adults (18+ subjects) were included, but justification for the upper age limit is needed.

Response 5: At that time, almost all people age over 60 had vaccinated by Thai government policy.

Point 6: It is impossible to conclude on "risk factors" in a cross-sectional design. At best, one might speak about "associations" (between certain subject characteristics and the fact of being seropositive) - but, when you only know 3-4 variables about the subjects, this makes no sense - if you only know age, gender, number of household members and broadly classified occupation - you cannot detect "independent association". And this analysis really does not add anything to this work. So, besides being conceptually wrong, it is non-informative - and should be removed.

Response 6: we removed table 4 and result on risk factors analysis.

Point 7: Obviously, it would have been too expensive to conduct all 25000+ tests using a commercial test. Therefore, authors developed an in-house immunoassay. This might be the "critical point" of the entire work. Method describes the technical part - but many other things have to be declared about this assay: (i) all analyses done at one place/lab? (ii) validation data for the assay ? (eg., "agreement" vs. a commercial one, intra-day and inter-day variability etc.). This is the critical bioanalytical procedure in this work - everything depends on it. Hence, it needs to be fully validated and this should be presented in the manuscript (or a Supplement). 

Response 7:  we added the validation data of in-house assay in 2.4.1.

Point 8:  A flow-chart should show how many tested positive in the "first-step" testing (in-hous assay) - i.e., how many were conveyed to further testing using a commercial assay (to yield the final "positives").

Response 8: we added flow chart in 3.2 Serological test result.

Point 9: Discussion should avoid wording like "representative sample"..."random sample." etc. It is true that subjects were enrolled without a "general prejudice" - they just needed to meet the key inclusion criterion and consent to provide a blood sample, but this does not mean that they are "random sample from the population".

Response 9: we have change representative sample to the meaning of samples that meet the key inclusion criterion and consent.

Point 10: Proportions of seroprevalence should not be interpreted in terms of "differences between regions" - but only in terms of "differences between samples from different regions". The sampling process is not appropriate for generalizations - hence, authors should be cautions in this respect

Response 10: we have changed in term of the Thai samples and describe not for compare in the 3.3 seroprevalence by geographic data.

Point 11: Overall, there are several key points that need to be clearly presented: (i) details of the sampling plan; (ii) validation data for the in-house assay; (iii) interpretation in terms of the "sample" -with very cautions reference to the "population of Thai adults who had not been vaccinated and/or had not experienced a clinically obvious COVID-19 episode".

Response 11:  we have added 2.2 sampling plan 2.4.1 detail of validation data for the in-house assay and rewrite in for sample instead of population.

Point 12: Authors should reconsider what is a "broader" interpretation of the findings. Do they really indicate what the last sentence of the Conclusions states?

Response 12: we have rewrited the conclusion.

Reviewer 2 Report

Title:

SARS- CoV-2 seroprevalence in unvaccinated adults in Thailand, November ,2021

The authors in this study survaied the unvaccinated  population of Thailand. The survey 26,717 people out of 12 public health area. Blood samples were collected and tested for anti CoV-2 antibodies. The authors used two methods to detect seroconvert samples.

  The first quantitative assay was  inhouse immunoassay (ELISA), in which  serum samples were captured with  RBD protein. The second assay used commercial assay ( Quant IgGII, Abbott Ireland). This assay was a chemiluminescent microparticles immunoassay quantifying samples against S1 CoV-2.

This cross sectional survey is very informative and important public health tool in mapping infections and unvaccinated population.

Comments:

1-      The authors used different immunosaay methods to process the samples. The first method tested the samples against RBD and the second tested against the whole spike protein.

The results should be processed separately and graphed with statistics.

2-      Positive samples in the inhouse kit  had an O.D ratio of 1.5 and above .

After 5 washer with 400 µl buffer per well, all the negative Samples should have a very low O.D ( eg. 0.5 – 0.8)

3-      The authors mentioned that the highest seropositive was in the BMI>30 kg/m2

According to table 4, there are 2,539 obese person out of 26717 and this will make 9.70% of the total samples and out of those 52 were positive which is 0.19% of all the samples.

Author Response

Response to Reviewer 2 Comments

Point 1: The authors used different immunoassay methods to process the samples. The first method tested the samples against RBD and the second tested against the whole spike protein. The results should be processed separately and graphed with statistics.

Response 1: we have added flow chart in 3.2 Serological test result.

Point 2: Positive samples in the in-house kit had an O.D ratio of 1.5 and above . After 5 washer with 400 µl buffer per well, all the negative Samples should have a very low O.D (e.g. 0.5 – 0.8)

Response 2: The cut-off ratio of 1.5 was determined as the mean values of OD ratio derived from confirmed negative samples plus two standard deviations, as described in 2.4.1

Point 3: The authors mentioned that the highest seropositive was in the BMI>30 kg/m2 According to table 4, there are 2,539 obese persons out of 26717 and this will make 9.70% of the total samples and out of those 52 were positive which is 0.19% of all the samples.

Response 3:  because of we would like to show the prevalence among BMI group so, it is 2.0% 52/2539 but if would like to show prevalence in BMI>30 kg/m2 cases in total samples it will be 0.19 (52/26717), however according to reviewer 1 the table 4 was removed.

Round 2

Reviewer 1 Report

The authors have improved the manuscript to a considerable extent.

English is still very poor.